# Impact of KIF4A on Cancer Stem Cells and EMT in Lung Cancer and Glioma

**DOI:** 10.3390/cancers15235523

**Published:** 2023-11-22

**Authors:** Yeon-Jee Kahm, In-Gyu Kim, Uhee Jung, Jei Ha Lee, Rae-Kwon Kim

**Affiliations:** 1Department of Environmental Safety Technology Research, Korea Atomic Energy Research Institute, Yuseong-Gu, Daejeon 34057, Republic of Korea; kahmyj@kaeri.re.kr (Y.-J.K.); igkim@kaeri.re.kr (I.-G.K.); uhjung@kaeri.re.kr (U.J.); 2Department of Radiation Life Science, Korea University of Science and Technology, Yuseong-Gu, Daejeon 34113, Republic of Korea; 3Department of Genetic Resources, National Marine Biodiversity Institute of Korea, Seocheon 33662, Republic of Korea; jeiha@mabik.re.kr

**Keywords:** kinesin family member 4A, cancer stem cells, epithelial-to-mesenchymal transition, plasminogen activator inhibitor-1

## Abstract

**Simple Summary:**

Kinesin family member 4A (KIF4A) is associated with poor prognosis in lung cancer and glioma. Our research found that KIF4A affects cancer stem cells (CSCs) and promotes epithelial-to-mesenchymal transition (EMT), a process linked to cancer’s aggressiveness and spread. We also identified its influence on the secretion of plasminogen activator inhibitor-1 (PAI-1), a key factor in cancer malignancy. KIF4A appears to sustain malignancy through an autocrine loop with PAI-1. These findings suggest that KIF4A has the potential to serve as a novel therapeutic target and prognostic marker in lung cancer and glioma, shedding light on mechanisms involved in cancer recurrence and metastasis.

**Abstract:**

Kinesin family member 4A (KIF4A) belongs to the kinesin 4 subfamily of kinesin-related proteins and is involved in the regulation of chromosome condensation and segregation during mitotic cell division. The expression of KIF4A in various types of cancer, including lung, breast, and colon cancer, has been found to be associated with poor prognosis in cancer patients. However, the exact mechanism by which it promotes tumorigenesis is not yet understood. In osteosarcoma, the expression of KIF4A has been shown to be associated with cancer stem cells (CSCs), whereas in breast cancer, it is not associated with the maintenance of CSCs but regulates the migratory ability of cells. In this light, we identified phenotypic phenomena affecting the malignancy of cancer in lung cancer and glioma, and investigated the mechanisms promoting tumorigenesis. As a result, we demonstrated that KIF4A affected lung cancer stem cells (LCSCs) and glioma stem cells (GSCs) and regulated CSC signaling mechanisms. In addition, the migratory ability of cells was regulated by KIF4A, and epithelial-to-mesenchymal transition (EMT) marker proteins were controlled. KIF4A regulated the expression of the secretory factor plasminogen activator inhibitor-1 (PAI-1), demonstrating that it sustains cancer malignancy through an autocrine loop. Taken together, these findings suggest that KIF4A regulates CSCs and EMT, which are involved in cancer recurrence and metastasis, indicating its potential value as a novel therapeutic target and prognostic marker in lung cancer and glioma.

## 1. Introduction

Lung cancer is responsible for more deaths than any other type of cancer [1]. Non-small cell lung cancer (NSCLC), a deadly advanced lung cancer, accounts for ~80% of all lung cancers and has a 5-year survival rate of only ~15% [2]. Meanwhile, glioblastoma (GBM) is the leading aggressive primary brain tumor and is still classified as an incurable brain disease despite the advent of surgery and various therapies [3]. GBM is difficult to treat with drugs due to the blood–brain barrier (BBB) and is primarily treated with surgery and radiation therapy, as well as some drug therapy [4,5]. The median survival time for patients with GBM is less than one year, and long-term survival is extremely rare [6]. The reasons for the poor prognosis of GBM patients are recurrence of the tumor after surgery, invasion to other parts of the body, and congenital or acquired resistance to chemotherapy and radiation therapy [7]. We previously analyzed gene expression in aldehyde dehydrogenase 1 (ALDH1)^+^ cells versus ALDH1^−^ cells [8]. Yan et al. cited the results of a gene analysis of CD133^+^ cells using 21 fresh GBM samples [9]. We selected genes that were increased in common between the two gene analysis results, one of which is kinesin family member 4A (KIF4A).

KIF4A belongs to the kinesin superfamily (KIFs), which are microtubule-based motor proteins that generate directional movement of various substances, including organelles, and protein complexes, along microtubules [10,11,12,13]. Kinesins are involved in many important cellular processes, including cell division. To date, 45 KIFs have been identified, and several of these families have demonstrated diverse functions in tumors [14]. It has recently emerged that KIF4A, a member of the kinesin superfamily, has a potential role in the malignant transformation of lung and brain cancers [15,16]. While KIF4A was initially identified for its roles in mitosis and cytokinesis, growing evidence suggests that it may contribute to the progression and aggressiveness of these malignancies [17,18,19].

In different types and stages of cancers, diverse cytokines are released from the tumor cells [20]. Some of these may act as tumor suppressors, but others as tumor promoters [21]. Above all, plasminogen activator inhibitor-1 (PAI-1), a member of the serine protease inhibitor (serpin) family, has emerged as a compelling candidate in the context of malignant transformation in lung and brain cancers [22,23]. Initially recognized for its role in the regulation of fibrinolysis, PAI-1 is now recognized as a multifunctional protein with diverse roles in cellular processes beyond its traditional antifibrinolytic function [24]. Accumulating evidence suggests that PAI-1 plays a significant role in cancer progression by promoting tumor growth, invasion, angiogenesis, and metastasis [25,26,27,28]. In lung cancer, elevated levels of PAI-1 have been observed in tumor tissues compared to normal lung tissue, and this upregulation has been associated with advanced disease stages, poor prognosis, and reduced survival rates [29,30]. Additionally, PAI-1 has been implicated in promoting resistance to chemotherapy and radiation therapy in lung cancer, further highlighting its clinical relevance [31,32].

In this study, we investigated the involvement of KIF4A in cancer stem cells (CSCs) of lung cancer and glioma, and the relationship between KIF4A and cytokine PAI-1. We found that overexpression of KIF4A in CSCs of lung cancer and glioma mediates tumor progression, invasion, and radiotherapy resistance. By exploring the molecular mechanisms of KIF4A in lung cancer stem cells and glioma stem cells, we aim to provide a comprehensive understanding of its role in these malignancies.

## 2. Materials and Methods

### 2.1. Cell Culture

This experiment used experimental methods and cell lines from a previously published paper [33]. We utilized the human lung carcinoma cell lines A-549 (CCL-185™) and NCI-H460 (HTB-177™), obtained from the American Type Culture Collection (ATCC^®^; Rockville, MD, USA). The cells were cultured in RPMI 1640 MEDIUM (Cat. No. SH30027.01) supplemented with 10% FBS (Cat. No. SH30919.03) and 1% Penicillin-Streptomycin Solution (Cat. No. SV30010) (Hyclone, Cytiva, Pittsburgh, PA, USA). The human glioblastoma cell line U-87 MG (Cat. No. 30014) was acquired from the Korea Cell Line Bank (KCLB, Seoul, Republic of Korea) and cultured in DMEM/HIGH GLUCOSE (Cat. No. SH30243.01), supplemented with 1% Penicillin-Streptomycin Solution (Cat. No. SV30010), and 10% FBS (Cat. No. SH30919.03) from Hyclone, Cytiva, Pittsburgh, PA, USA. Both cell lines were maintained in an incubator at 37 °C with 5% CO_2_.

### 2.2. Sphere-Forming Media Treatment

DMEM/F12 (Cat. No. 11320-033; Invitrogen, Waltham, MA, USA), supplemented with B27 Serum-Free Supplement (Cat. No. 17504-044; Invitrogen, Waltham, MA, USA), basic Fibroblast Growth Factor (bFGF, Cat. No. 13256-029; Invitrogen, Waltham, MA, USA), Epidermal Growth Factor (EGF, Cat. No. E9644; Sigma-Aldrich, Burlington, VT, USA), N-2 supplement (Cat. No. 17502048; Invitrogen, Waltham, MA, USA), and 1% Penicillin-Streptomycin Solution (Cat. No. SV30010; Hyclone, Cytiva, Pittsburgh, PA, USA) was used as the sphere-forming media. After the treatment of the sphere-forming media for at least three days, A-549 (CCL-185™) and U-87 MG (30014) were changed into CSCs.

### 2.3. siRNA Transfection

To knockdown KIF4A expression in A-549 (CCL-185™) and U-87 MG (30014) cell lines, we employed the small interfering RNA (siRNA) transfection method. The siRNA specific for targeting KIF4A (sequence: AGUUGACUCGACUGCUUCA, UGAAGCAGUCGAGUCAACU) was obtained from Bioneer Corporation, Daejeon, Republic of Korea. For transfection, 10 pmol of siKIF4A was introduced using Lipofectamine RNAi MAX reagent (Cat. No. 13-778-150; Invitrogen, Waltham, MA, USA). As a negative control, Stealth RNAi Negative Control Medium GC (Cat. No. 12935-300; Invitrogen, Waltham, MA, USA) was utilized. Following siKIF4A treatment, cells were incubated at 37 °C with 5% CO_2_ for 72 h in an incubator.

### 2.4. Western Blot

In this experiment, we used experimental methods and reagents from previously published papers [33]. To prepare cell samples, RIPA Lysis Buffer (Cat. No. 20-188; Millipore, Burlington, VT, USA) supplemented with phosphatase inhibitor cocktail tablets (Cat. No. 04906837001) and protease inhibitor cocktail tablets (Cat. No. 1836153001) from Roche (Basel, Switzerland) was utilized for cell lysis. Approximately 40 µg of proteins were prepared as loading samples and separated on 8–12% sodium dodecyl sulfate (SDS)-polyacrylamide gels. Subsequently, the separated proteins were transferred onto Amersham™ Protran™ 0.2 µm NC membranes (Cat. No. 10600001; Amersham™, Cytiva, Pittsburgh, PA, USA). Phosphate-buffered saline containing 10% nonfat milk served as the blocking buffer. Following a 30-min blocking step, the membranes were incubated with primary antibodies overnight at 4 °C. On the following day, we washed the membranes with Tris-buffered saline (TBS) buffer (Cat. No. A0027), supplemented with 0.1% Tween 20 (Cat. No. TB0560) from BIO BASIC in Markham, Canada. HRP-linked secondary antibodies, either anti-rabbit IgG (Cat. No. 7074S) or anti-mouse IgG (Cat. No. 7076S) from Cell Signaling Technology, Inc. in Danvers, MA, USA, were then applied. After a one-hour incubation with the secondary antibodies, the membranes were rinsed and visualized using Western blotting luminol reagent (Cat. No. sc-2048) from Santa Cruz Biotechnology, Inc. in Dallas, TX, USA. Densitometric analysis was carried out to quantify protein expression levels utilizing the image processing program ImageJ (imagej.net/ij/index.html, accessed on 13 June 2022). The original western blot figures of Appendix A can be found in Appendix A.

### 2.5. PCR

This experiment was conducted using the same experimental method and primers used in the published paper [33]. cDNA was synthesized from 1 μg of RNA extracted from A-549 and U-87 MG cell lines using Maxime RT Premix (Random Primers) (Cat. No. 25082). Maxime™ PCT PreMix (i-Taq) (Cat. No. 25026) (iNtRON Biotechnology, Seongnam, Republic of Korea) was added with 1 µL of cDNA and 19 µL of DEPC water. PCR was conducted using a T100 thermal cycler from Bio-Rad Laboratories, Inc., based in Hercules, CA, USA. The primer sequences and experimental conditions used in the PCR experiment are summarized in Table 1.

### 2.6. Immunocytochemistry

In this experiment, we used experimental methods and antibodies from previously published papers [33]. Cells were seeded with the A-549 and U-87 MG cell lines at a density of 1 × 10^5^ in 35 mm cell culture plates using cover glass. To achieve fixation, we utilized 4% paraformaldehyde (Cat. No. P2031; Biosesang, Seongnam, Republic of Korea). The fixed cells were washed with phosphate-buffered saline (PBS) and incubated for one day within each primary antibody. The next day, after washing the samples with ICC Blocking Solution (made of PBS 44 mL, FBS 5 mL, and Tween 20 1 mL), secondary antibodies (Alexa Fluor™ 488 donkey anti-mouse IgG (Cat. No. A21202; Invitrogen, Waltham, MA, USA) and Alexa Fluor™ 488 donkey anti-rabbit IgG (Cat. No. A21206; Invitrogen, Waltham, MA, USA)) were used. The nuclei were stained by DAPI solution (Cat. No. sc-24941; Santa Cruz Biotechnology, Inc., Dallas, TX, USA) and a Zeiss LSM510 meta-microscope (Carl Zeiss Microimaging GmbH, Göttingen, Germany) were used for imaging the samples.

### 2.7. Single Cell Assay

Each well of 96 well plates (Cat. No. 3474; Corning, Inc., Corning, NY, USA) received a single cell, and these cells were subsequently cultured in sphere-forming media. These plates were incubated for at least seven days in 5% CO_2_ with 37 °C conditions. After 7–10 days, the number of spheroids formed in each well were counted. 

### 2.8. Limited Dilution Assay

A total of 1, 10, 50, 100, 150, and 200 cells were seeded in each 96 well plate (Cat. No. 3474; Corning, Inc., Corning, NY, USA) and incubated in 5% CO_2_ at 37 °C for at least 7 days. After incubation for at least seven days, only the number of wells in which cells had formed spheroids were counted.

### 2.9. Colony Formation Assay and Irradiation

In this experiment, we used the experimental method and reagent from a previously published paper [33]. Initially, 35 mm cell culture plates (Cat. No. 430165; Corning, Inc., Corning, NY, USA) were seeded with 1 × 10^3^ cells, then incubated for one day at 37 °C in a 5% CO_2_ environment. On the following day, the cells were exposed to 3 Gy γ-irradiation (Korea Atomic Energy Research Institute) and kept in a 5% CO_2_ atmosphere at 37 °C for a duration of 7–10 days. Afterward, the cell colonies were stained using 1 mL of crystal violet (Cat. No. HT90132; Sigma-Aldrich; Burlington, VT, USA).

### 2.10. Wound Healing Assay

In this experiment, we used experimental methods and reagents from a previously published paper [33]. Cells were initially seeded at a density of 2 × 10^5^ in 35 mm cell culture plates (Cat. No. 430165; Corning, Inc., Corning, NY, USA) and allowed to incubate for one day. When the cell population reached 80% confluence, a 200 μL micropipette tip was employed to create a controlled scratch in the monolayer of the cultured cells. The suspended cells were washed with PBS and the plates were then incubated with RPMI 1640 medium (Cat. No. SH30027.01; Hyclone; Cytiva, Pittsburgh, PA, USA) supplemented with 10% FBS (Cat. No. SH30919.03; Hyclone; Cytiva, Pittsburgh, PA, USA) and 1% Penicillin-Streptomycin Solution (Cat. No. SV30010; Hyclone; Cytiva, Pittsburgh, PA, USA).

### 2.11. Invasion and Migration Assays

In this experiment, we used experimental methods and reagents from a previously published paper [33]. We used 6.5 mm Transwell^®^ inserts with an 8.0 µm pore polycarbonate membrane (Cat. No. 3422; Corning, Inc., Corning, NY, USA). The target cells were initially seeded in the upper chamber using 150 μL of Opti-MEM^®^ (serum-free media, Cat. No. 31985-070; Invitrogen, Waltham, MA, USA). In the lower chamber, we added 800 μL of RPMI 1640 medium (Cat. No. SH30027.01) or DMEM/HIGH GLUCOSE (Cat. No. SH30243.01) from Hyclone, Cytiva, Pittsburgh, PA, USA, both supplemented with 10% FBS and 1% Penicillin-Streptomycin Solution. Especially for the invasion assay, the upper chamber was coated with 200 μL of Matrigel. After incubating the samples at 37 °C with 5% CO_2_ for one day, cells that migrated and invaded from the upper chamber to the bottom of the chamber were stained with crystal violet (Cat. No. HT90132; Sigma-Aldrich; Merck KGaA; Burlington, VT, USA).

### 2.12. Cytokine Array

In this experiment, we used experimental methods and reagents from a previously published paper [33]. To assess the cytokines released from the cells, we employed a Human Cytokine Array Kit (Catalog No. ARY005B; R&D SYSTEMS, Minneapolis, MN, USA). The cultured media were collected for cytokine array analysis. Following membrane blocking for one hour, 15 μL of reconstituted human cytokine array detection antibody cocktail was added to the prepared cultured media. Subsequently, the mixture was incubated for one hour at 10–20 °C. The prepared sample was treated on the array membrane after blocking. The day after incubation, each membrane was washed and visualized using a chemical reagent mixture. 

### 2.13. Antibodies

For Western blot and immunocytochemistry, CD133 (Cat. No. ab1998), ALDH1A1 (Cat. No. ab6192), ALDH1A3 (Cat. No. ab129815), E-cadherin (Cat. No. ab15148) (Abcam, Cambridge, UK), KIF4 (Cat. No. sc-365144), β-actin (Cat. No. sc-47778), SNAIL (Cat. No. sc-10432), SLUG (Cat. No. sc-166476), Twist (Cat. No. sc-15393), ZEB1 (Cat. No. sc-25388) (Santa Cruz Biotechnology, Inc., Dallas, TX, USA), CD44 (Cat. No. 3570), Sox2 (Cat. No. 3579), Oct-4 (Cat. No. 2750), Nanog (Cat. No. 4893) (Cell Signaling Technology, Inc., Danvers, MA, USA), N-cadherin (Cat. No. 610920) (BD Transduction, Franklin Lakes, New Jersey, NJ, USA), and vimentin (Cat. No. MA5-14564) (Invitrogen; Thermo Fisher Scientific, Inc., Waltham, MA, USA). 

### 2.14. Kaplan–Meier Survival Analysis

We used a Kaplan–Meier survival graph, published genetic information system (https://kmplot.com/analysis/, accessed on 13 June 2022), to obtain lung cancer patients’ survival graph according to the level of KIF4A gene expression. A total of 2166 patients of lung cancer were analyzed to plot the survival graph.

### 2.15. Statistical Analysis

All experiments were conducted with a minimum of three independent repetitions. The results are expressed as mean ± standard deviation, and the specific *p*-values for each comparison are detailed in the corresponding figure legends. To validate the data, all graphs underwent a two-sided paired Mann–Whitney U analysis. Statistical significance was defined as a *p*-value less than 0.05.

## 3. Results

### 3.1. KIF4A Is Highly Expressed in Malignant Lung Cancer Cells

Based on our previous findings and the work of other groups, we sought to find new target proteins [8,9]. We selected a group of genes that are highly expressed in ALDH1^+^ lung cancer cells and CD133^+^ glioma cells, and identified the function of the KIF4 gene. We first looked at the expression of KIF4A in CSCs, where culturing cells in sphere-forming medium increases CSC marker proteins (Figure 1A). In addition, the protein expression of KIF4A was increased in the sphere-forming-media-treated group. The CSC marker proteins we compared were ALDH1A1, ALDH1A3, and CD133, and the results showed that all CSC marker proteins and KIF4 were highly expressed in the sphere-forming-media-treated group. Figure 1B compares the expression of CSC marker mRNA with that of KIF4 mRNA, which showed the same trend as the protein expression levels. The expression of CSC marker proteins and KIF4A in the sphere-forming-media-treated group was visually confirmed using ICC (Figure 1C). The CSC marker proteins ALDH1A1 and CD133 were both highly expressed in the sphere-forming-media-treated group, and KIF4A was also highly expressed in the sphere-forming-media-treated cells. In addition, the Kaplan–Meier plot of lung cancer patients showed that high expression of KIF4A was associated with poor prognosis (Figure 1D).

### 3.2. KIF4A Regulates the Properties of Lung Cancer Stem Cells

CSCs exhibit the ability to form spheres in sphere-forming media. Upon suppression of KIF4A expression, we evaluated the sphere-forming capability under sphere-forming media conditions (Figure 2A). The results revealed a significant reduction in sphere-forming ability in the KIF4A gene-suppressed group. CSCs are also known for their self-renewal capacity, which we assessed through a single-cell assay (Figure 2B). The reduction in KIF4A led to a decrease in self-renewal capacity. Additionally, we conducted a limited dilution assay to further measure the self-renewal capacity of CSCs, and the results indicated a significant reduction in spherical formation ability in the group where gene expression was reduced using siRNA (Figure 2C). To investigate whether KIF4A is involved in the regulation of CSCs, we analyzed the expression of CSC marker proteins after KIF4A expression was knocked down using siRNA. The results demonstrated significant suppression in the expression of ALDH1A1, ALDH1A3, and CD44 (Figure 2D). Figure 2E presents the results of an analysis on whether the expression of CSC regulators (SOX2, Oct-4, Nanog), known to play pivotal roles in CSCs, is regulated by KIF4A expression. The expression of CSC regulators was diminished in the group where KIF4A gene expression was suppressed. To visually confirm the changes in CSC marker proteins, we conducted experiments utilizing the immunocytochemistry (ICC) method (Figure 2F). In the group where KIF4A expression was suppressed, the expression of CSC marker proteins was similarly diminished. Furthermore, when A549 cells were exposed to 3 Gy of radiation, more than half of them survived. However, in the group where KIF4A expression was reduced by siRNA, the extent of colony formation was notably decreased (Figure 2G). We also conducted experiments using H460 cells, which have lower KIF4A expression than A549 cells, to verify that KIF4A regulates the properties of lung cancer stem cells (Appendix A). As a result, we observed that KIF4A overexpression also enhanced the expression of CSCs markers in this lung cancer cell line.

### 3.3. The Epithelial Mesenchymal Transition Is Regulated by KIF4A

Many studies have shown that cells with CSC characteristics are involved in epithelial-to-mesenchymal transition (EMT). Therefore, we conducted experiments to investigate whether KIF4A is involved in the regulation of EMT. When KIF4 expression was suppressed using siRNA, E-cadherin, N-cadherin, and vimentin, all known EMT marker proteins, were altered. E-cadherin is an epithelial marker protein and was increased in the group where the expression of KIF4A was suppressed, while the mesenchymal marker proteins N-cadherin and vimentin were decreased (Figure 3A). The expression of Snail, Slug, TWIST, and ZEB1, known as EMT regulatory proteins, was significantly reduced in the siKIF4A treatment group (Figure 3B). To confirm this visually, the expression levels of EMT marker proteins were analyzed by the ICC method (Figure 3C). As shown in Figure 3A, changes in EMT marker proteins were observed in the si-KIF4 treatment group. The EMT phenomenon is related to the migratory ability of cells. Suppressing the expression of KIF4A using siRNA significantly reduced the migratory ability of cells (Figure 3D). KIF4A regulates cell migration and invasion (Figure 3E), which are important factors in cancer metastasis. The number of migrating and invading cells was significantly reduced in the siKIF4A treated cell lines. We conducted an experiment using H460 cells, which have poor motility, to determine the extent to which KIF4A is involved in EMT. Our findings revealed that H460 cells exhibited lower migratory potential when compared to A549 cells. To assess the impact of KIF4A overexpression, we transfected H460 cells with KIF4A and subsequently measured their migratory and invasive capabilities (Appendix A). As a result, it was confirmed that cell motility was increased in H460 with overexpressed KIF4A.

### 3.4. Identification and Functional Analysis of Cytokines Regulated by KIF4A

Cytokines are involved in the malignant transformation of cancer and are one of the factors that hinder cancer treatment [33]. We conducted experiments to identify cytokines that are regulated by KIF4A. Cytokine array kits were used to identify cytokines regulated by the expression of KIF4A (Figure 4A). The secretion of CXCL1, IL-8, MIF, and PAI-1 was reduced in the group that suppressed KIF4A expression. To validate the results of the cytokine array kit, changes in mRNA were measured using qPCR (Figure 4B). The results showed that the RNA of all cytokines with reduced secretion was reduced. Experiments were conducted to analyze the effect of the decreased secretion of cytokines on the malignancy of cancer. First, a sphere-forming assay was performed to check the sphere-forming ability of the cells (Figure 4C). The group treated with the PAI-1 neutralizing antibody showed the greatest reduction in sphere formation ability. Next, the migration and invasion capacity of cells was analyzed using a Boyden chamber (Figure 4D). Each chamber was treated with a neutralizing antibody capable of inhibiting each cytokine. The results showed that the migration and invasion ability of the cells was most significantly affected by PAI-1, consistent with the sphere formation results. To investigate the correlation between secreted PAI-1 and KIF4A, we analyzed the protein expression of KIF4A after treatment with a PAI-1 neutralizing antibody (Figure 4E). We also checked whether the gene expression of KIF4A was affected by a PAI-1 neutralizing antibody, and the results were consistent with those in Figure 4E (Figure 4F). The results showed that the protein expression of KIF4A was reduced by PAI-1, and the expression of PAI-1 was also reduced. To determine if CSCs are regulated by PAI-1, cells were treated with a PAI-1 neutralizing antibody (Figure 4G). CD44, ALDH1A1, and ALDH1A3, markers of CSCs, were all decreased in cells treated with a PAI-1 neutralizing antibody. E-cadherin, a marker of epithelium, was increased, while N-cadherin and vimentin, markers of mesenchyme, were decreased (Figure 4H). 

### 3.5. Functional Analysis of KIF4A in Glioma Stem Cells

KIF4A is a gene that we identified in ALDH1^+^ lung cancer cells and CD133^+^ glioma cells. Subsequently, we conducted experiments to elucidate the function of KIF4A in glioma stem cells. We treated U87 cells, a glioma cell line, with sphere-forming media and observed that the expression of CSC marker proteins, including CD133, ALDH1A1, and ALDH1A3, was diminished in the group where KIF4A expression was knocked down (Figure 5A). Visual confirmation through ICC further supported these findings, as shown in Figure 5B. To evaluate the ability to form spheres, a hallmark of CSCs, we conducted a sphere-forming assay (Figure 5C). The results indicated a significant reduction in sphere formation in the group where KIF4A was inhibited using siRNA. In an effort to investigate whether KIF4A is involved in EMT, we examined changes in EMT marker proteins (Figure 5D). The expression of an epithelial marker protein, E-cadherin, increased, while mesenchymal marker proteins, N-cadherin and vimentin, decreased in the group where KIF4A expression was inhibited. Further comparison of the expression of each EMT marker protein through ICC is shown in Figure 5E. To assess the migration and invasion abilities of cells, we performed experiments using Boyden chambers (Figure 5F). The results revealed reduced migration and invasion abilities in cells with low expression of KIF4A. To verify the function of KIF4A in glioma cells in CSC and EMT, we additionally performed the experiments using siRNA treated U373 cells, and observed that KIF4A regulates CSC and EMT properties not only in U87 cells but also in U373 cells (Appendix A). We also investigated the expression of PAI-1, a gene regulated by KIF4A, in a glioma cell line (Figure 5G). The expression of PAI-1 was found to be decreased in cells where KIF4A expression was inhibited by treatment with siRNA.

## 4. Discussion

In this study, we investigated the role of KIF4A in lung cancer and glioma, focusing on its involvement in CSC and EMT, which play critical roles in tumor progression, metastasis, and treatment resistance. Our findings shed light on the potential of KIF4A as a therapeutic target and prognostic marker in these malignancies.

We first demonstrated that KIF4A is highly expressed in malignant lung cancer cells and that its expression correlates with poor prognosis in lung cancer patients. The direct relationship with lung cancer patients will be explored in further experiments. To elucidate the functional significance of KIF4A in LCSCs, we knocked down the expression of KIF4A and evaluated the effects on CSC properties. These results suggest that KIF4A plays an important role in maintaining stemness and the self-renewal potential of mesenchymal stem cells. We also investigated the involvement of KIF4A in EMT, a process associated with increased invasiveness and metastatic potential of cancer cells. Inhibition of KIF4A affected EMT marker proteins (E-cadherin, N-cadherin, vimentin) and EMT regulatory proteins (Snail, Slug, TWIST, ZEB1). These results suggest that KIF4A plays a role in promoting the mesenchymal phenotype and migratory ability of cancer cells through the regulation of EMT.

To gain insight into the mechanisms underlying the effects of KIF4A, we investigated the cytokines regulated by KIF4A expression. A cytokine array analysis showed that knockdown of KIF4A reduced the secretion of CXCL1, IL-8, MIF, and PAI-1. Among these cytokines, PAI-1 was shown to have the greatest impact on cancer malignancy due to its role in sphere formation, migration, and invasion of cancer cells. Notably, we observed cross-regulation between KIF4A and PAI-1, suggesting the existence of an autoregulatory loop involving KIF4A and PAI-1 in maintaining cancer malignancy. We plan to further explore this signaling mechanism. PAI-1 is a serine protease inhibitor that has been implicated in various aspects of cancer progression, including tumor growth, invasion, angiogenesis, metastasis, and therapy resistance. Our findings demonstrate a significant reduction in PAI-1 secretion upon suppression of KIF4A expression in lung cancer and glioma cells. Elevated levels of PAI-1 have been observed in lung tumor tissues compared to normal lung tissue, and this upregulation has been associated with advanced disease stage, poor prognosis, and reduced survival rates [29,30]. Furthermore, PAI-1 has been implicated in promoting resistance to chemotherapy and radiation therapy in lung cancer, further emphasizing its clinical relevance [31,32]. The involvement of PAI-1 in cancer progression has been extensively studied, and dysregulation of PAI-1 in various cancer types, including lung cancer and glioma, is associated with poor prognosis. Increased expression of PAI-1 correlates with increased invasiveness, angiogenesis and metastasis, and resistance to therapy [31,32]. In addition, PAI-1 has been identified as a potential therapeutic target in lung cancer and glioma. Therefore, the identification of KIF4A as a regulator of PAI-1 provides valuable insight into the molecular mechanisms underlying cancer progression and emphasizes the potential of both KIF4A and PAI-1 as novel therapeutic targets in lung cancer and glioma.

## 5. Conclusions

In conclusion, our study highlights the critical role of KIF4A in promoting cancer stem cells and epithelial-to-mesenchymal transition, contributing to the aggressiveness and metastatic potential of lung cancer and glioma. Furthermore, we found out that KIF4A is a key regulator of PAI-1 which is secreted from lung cancer and glioma. These findings offer valuable insights into the molecular mechanisms underlying cancer progression and underscore the potential of both KIF4A and PAI-1 as promising therapeutic targets in lung cancer and glioma. Further research is warranted to explore the clinical implications and therapeutic opportunities associated with targeting KIF4A and PAI-1 in these malignancies.

## Figures and Tables

**Figure 1 cancers-15-05523-f001:**
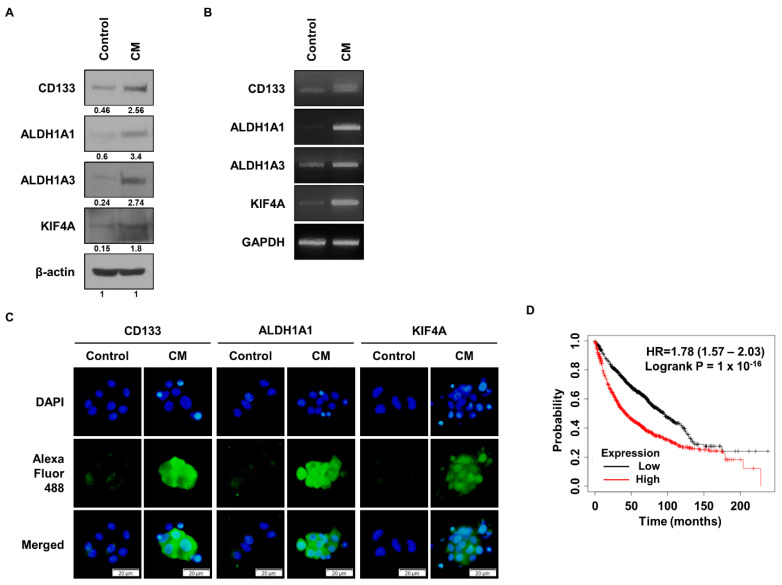
Expression analysis of KIF4A and cancer stem cell markers after sphere-forming media treatment. (**A**) Western blot analysis of the expression of KIF4A along with CD133, ALDH1A1, and ALDH1A3, marker proteins of CSCs, after sphere-forming media treatment in A549 cells. The numbers indicated beneath the blots represent the expression density of each protein, relative to β-actin. (**B**) Comparison of the expression of CSC marker genes and KIF4A after sphere-forming media treatment. (**C**) Analysis of the expression level of CSC marker proteins and KIF4A using immunocytochemistry. (**D**) Kaplan–Meier survival graph of lung cancer patients according to the level of KIF4A gene expression. Statistical significance was determined using log-rank tests. The original western blot figures can be found in Appendix A.

**Figure 2 cancers-15-05523-f002:**
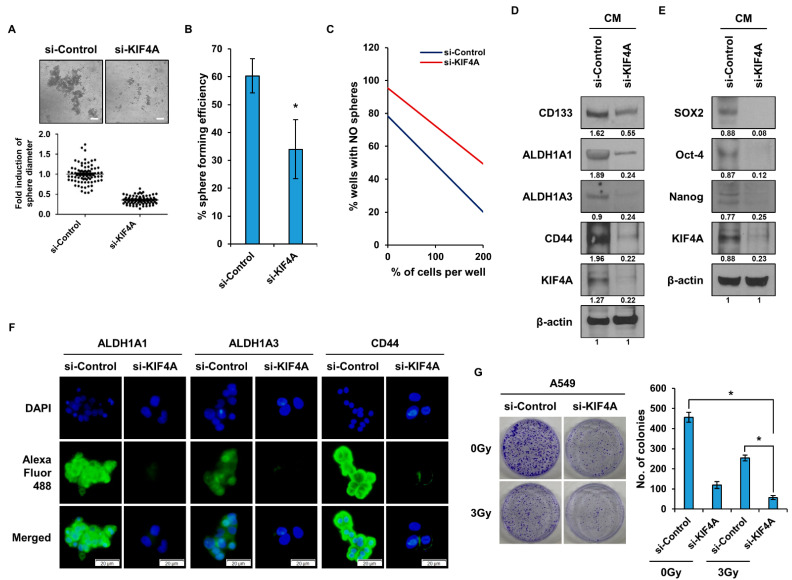
Investigation of KIF4A-mediated effects on cancer stem cells. (**A**) A sphere-forming assay was performed by treating A549 cells with siKIF4A followed by sphere-forming media. After sphere-forming media treatment, the cells were cultured for 10 days. (**B**) A549 cells treated with siRNA were seeded in a 96-well plate, and after 10 days, wells containing cells in which sphere formation occurred were counted. (**C**) Seeding by serial dilution of siRNA-treated A549 was performed in 96-well plates. After seven days, only wells with sphere-forming cells were counted. (**D**) A549 cells in which KIF4A expression was suppressed were treated with sphere-forming media, and changes in CSC marker proteins were analyzed using Western blotting. The numbers indicated beneath the blots represent the expression density of each protein, relative to β-actin. (**E**) A549 cells in which KIF4A expression was suppressed were treated with sphere-forming media, and changes in CSC regulatory proteins were analyzed. The numbers indicated beneath the blots represent the expression density of each protein, relative to β-actin. (**F**) The expression of the CSC marker protein was visualized according to the level of KIF4A gene expression using the ICC analysis. (**G**) Comparative analysis of resistance ability to irradiation according to the expression level of KIF4A. Data are presented as the mean ± standard deviation of three independent experiments. Scale bar, 50 µm. * *p* ≤ 0.05. The original western blot figures can be found in Appendix A.

**Figure 3 cancers-15-05523-f003:**
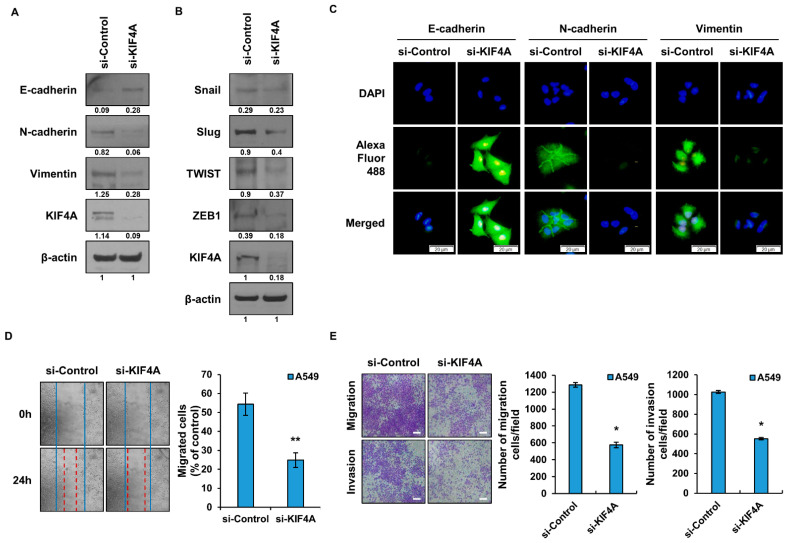
Exploring the role of KIF4A in EMT and cellular behavior of lung cancer cells. (**A**) Changes in EMT marker proteins according to KIF4A expression in A549 cells were analyzed by the WB method. The numbers indicated beneath the blots represent the expression density of each protein, relative to β-actin. (**B**) Analysis of changes in EMT regulatory proteins in A549 cells treated with siRNA. The numbers indicated beneath the blots represent the expression density of each protein, relative to β-actin. (**C**) EMT marker proteins were visually expressed using an ICC analysis. (**D**) A549 cells were wounded and the degree of filling of the area was compared and analyzed. (**E**) Cell migration and invasion abilities were analyzed using a Boyden chamber. Data are presented as the mean ± standard deviation of three independent experiments. Scale bar, 50 µm. * *p* ≤ 0.05, ** *p* ≤ 0.005. The original western blot figures can be found in Appendix A.

**Figure 4 cancers-15-05523-f004:**
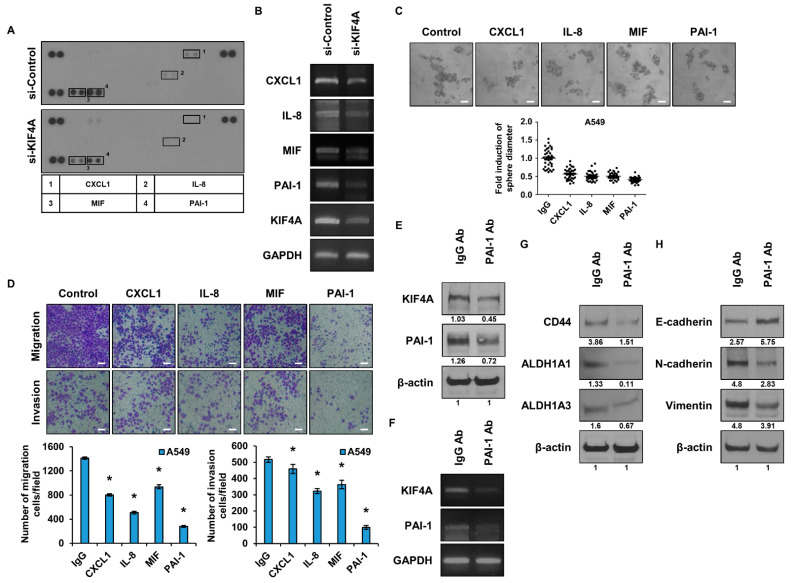
Regulatory role of KIF4A in cytokine expression within cancer stem cells. (**A**) The expression of KIF4A was suppressed using a cytokine array kit, and cytokines were screened. (**B**) RNA changes in cytokines were analyzed using qPCR. (**C**) After treating A549 cells with each neutralizing antibody, a sphere-forming assay was performed. (**D**) After each neutralizing antibody was treated with A549 cells, migration and invasion abilities were analyzed using a Boyden chamber. (**E**) After treatment with PAI-1 neutralizing antibody, the protein expression levels of KIF4A and PAI-1 were analyzed using WB. The numbers indicated beneath the blots represent the expression density of each protein, relative to β-actin. (**F**) After treatment with a PAI-1 neutralizing antibody, RNA expression levels of KIF4A and PAI-1 were analyzed using qPCR. (**G**) After treatment with a PAI-1 neutralizing antibody, expression levels of CSC marker proteins were analyzed by WB. The numbers indicated beneath the blots represent the expression density of each protein, relative to β-actin. (**H**) After treatment with a PAI-1 neutralizing antibody, expression levels of EMT marker proteins were analyzed by WB. The numbers indicated beneath the blots represent the expression density of each protein, relative to β-actin. Data are presented as the mean ± standard deviation of three independent experiments. Scale bar, 50 µm. * *p* ≤ 0.05. The original western blot figures can be found in Appendix A.

**Figure 5 cancers-15-05523-f005:**
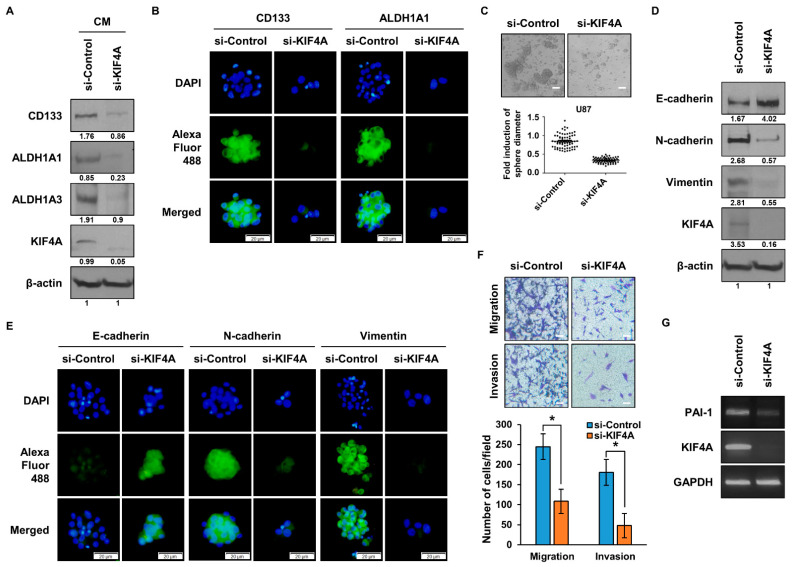
KIF4A regulates cancer stem cells, EMT, and cell behavior in glioma. (**A**) U87 cells were treated with siRNA, then cultured in sphere-forming media, and expression levels of CSC marker proteins (CD133, ALDH1A1, ALDH1A3) were compared and analyzed. The numbers indicated beneath the blots represent the expression density of each protein, relative to β-actin. (**B**) U87 cells were treated with siRNA and cultured in sphere-forming media, and the expression levels of CSC marker proteins were compared and analyzed using ICC. (**C**) The sphere-forming ability of U87 cells treated with siRNA was measured. (**D**) The expression of EMT marker proteins (E-cadherin, N-cadherin, and vimentin) in U87 cells treated with siRNA was compared and analyzed. The numbers indicated beneath the blots represent the expression density of each protein, relative to β-actin. (**E**) The expression of EMT marker proteins in siRNA-treated U87 cells was comparatively analyzed using ICC. (**F**) When the expression of KIF4A was inhibited in U87 cells, cell migration and invasion abilities were analyzed using a Boyden chamber. (**G**) After inhibiting the expression of KIF4A in U87 cells, the gene expression of PAI-1 was analyzed by qPCR. Data are presented as the mean ± standard deviation of three repeats. Scale bar, 50 µm. * *p* ≤ 0.05. The original western blot figures can be found in Appendix A.

**Table 1 cancers-15-05523-t001:** PCR primer sequences and experimental conditions.

Gene Name	Primer Sequence (5′ to 3′)	°C	Cycle
ALDH1A1	F: TTAGCAGGCTGCATCAAAAC	56	34
	R: GCACTGGTCCAAAAATCTCC		
ALDH1A3	F: ACCTGGAGGGCTGTATTAGA	57.5	34
	R: GGTTGAAGAACACTCCCTGA		
CD133	F: CATGGCCCATCGCACT	55	34
	R: TCTCAAAGTATCTGG		
CXCL1	F: ATGGCCCGCGCTGCTCTCTC	56	34
	R: TCAGTTGGATTTGTCACTGTTC		
GAPDH	F: AGTCAACGGATTTGGTCGTA	56	34
	R: GTCATGAGTCCTTCCACGAT		
IL-8	F: ATGGCTGCTCAAGGCTGGTC	56	34
	R: AGGCTTTTCATGCTCAACACTAT		
KIF4A	F: GAATAAAGCGCTTGCACTGA	55.5	34
	R: ACCACGCACTTCAGTAAGG		
MIF	F: GCGCGTGCGTCTGTGCC	56.5	34
	R: GACCACGTGCACCGCGATGTA		
PAI-1	F: CAGGCGGACTTCTCCAGTT	56.5	34
	R: CATTCGGGCTGAGACTACAAG		

## Data Availability

The data presented in this study are available in this article.

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
