# Peer review of "Impact of KIF4A on Cancer Stem Cells and EMT in Lung Cancer and Glioma"

_cancers, 2023, doi:10.3390/cancers15235523_

Round 1
Reviewer 1 Report
Comments and Suggestions for Authors
The authors present a compact descriptive paper describing the impact of KIF4A knockout on stem cells and EMT in in vitro lung and glioma cancer cell lines.
Altough having the exact same structure as the previous publications by the group and showing no clinical or translational data, the paper is a comprehensive analysis of KIF4A knockout effects on cancer cell lines. The data is well structured, some of the results are novel.
Minor complaints:
The english level is good, but smaller mistakes should be adressed.
Comments on the Quality of English LanguageThe english level is good with minor mistakes
Reviewer 2 Report
Comments and Suggestions for Authors
Introduction: Should be improved. There is a sudden shift from KIF4 to Plasminogen activator inhibitor-1 (PAI-1). Authors simply write that they explored mechanism of KIF4A mediated tumor progression or invasion. They should rather discuss objectives and findings briefly.
Methodology:
Invasion and migration assays: Authors have mentioned a common list of reagents and protocol that was followed for measuring migration and invasion. Did they use Matrigel coated inserts for invasion assay? Please mention what was the difference between these two approaches that were followed in this study.
Comment1: Authors should perform densiometric analysis to quantify quantify the level of KIF4A and other protein's overexpression shown in the blot. It can be a marginal alteration. Ex. in the blot ALDH1A3 expression looks almost same in control vs CM. Blot for KIF4A should be replaced it is not convincing.
Comment2: Fig 1 C: This is in general comment, throughout the manuscript wherever authors have shown protein expression in IHC they should show brightfield image panel which can show cells not positive for antibody staining. Empty channels (Alexa 488) in for control condition still needs validation that there are unstained cells in the panel.
Comment3: Fig. 1 D: How many number of samples were analyzed to plot survival graph? There is no mention from where does author used the dataset was it acquired from online data depository, if yes, please mention which dataset has been used in this analysis.
Comment4: Authors check KIF4 mediated regulation of EMT. They should compare levels of KIF4A in metastatic A549 cell line with any other lung cancer cell line model which is epithelial and has not gone metastatic transition. This will clearly explain how levels of KIF4A regulates EMT.
Comment 5: Fig. 5A, D: In the representative blots knockdown levels of KIF4A are strikingly different compared to similar loading control, actin. Does it mean in U87 cell line there is variation in knockdown efficiency of KIF4A if yes, it will be difficult to interpret the experimental results in figure 5. Please provide explanation.
Comments on the Quality of English Language
Line 278: Typo "Sanil" should be replaced by "Snail".
Round 2
Reviewer 2 Report
Comments and Suggestions for Authors
Manuscript in the revised version seems to be well improved.
